# Extracellular Vesicles from M1-Polarized Macrophages Combined with Hyaluronic Acid and a β-Blocker Potentiate Doxorubicin’s Antitumor Activity by Downregulating Tumor-Associated Macrophages in Breast Cancer

**DOI:** 10.3390/pharmaceutics14051068

**Published:** 2022-05-17

**Authors:** Carla Jorquera-Cordero, Pablo Lara, Luis J. Cruz, Timo Schomann, Anna van Hofslot, Thaís Gomes de Carvalho, Paulo Marcos Da Matta Guedes, Laura Creemers, Roman I. Koning, Alan B. Chan, Raimundo Fernandes de Araujo Junior

**Affiliations:** 1Department of Orthopedics, University Medical Center Utrecht, 3584 CX Utrecht, The Netherlands; c.a.jorqueracordero@umcutrecht.nl (C.J.-C.); l.b.creemers@umcutrecht.nl (L.C.); alanchan@clara.net (A.B.C.); 2Percuros B.V., 2333 CL Leiden, The Netherlands; tschomann@percuros.com (T.S.); thaisbida2013@gmail.com (T.G.d.C.); 3Translational Nanobiomaterials and Imaging (TNI) Group, Radiology Department, Leiden University Medical Center, Albinusdreef 2, 2333 ZA Leiden, The Netherlands; l.j.cruz_ricondo@lumc.nl (L.J.C.); a.vanhofslot@gmail.com (A.v.H.); 4Postgraduate Program in Health Science, Health Science Department, Federal University of Rio Grande do Norte (UFRN), Natal 59078 970, RN, Brazil; 5Cancer and Inflammation Research Laboratory, Department of Morphology, Federal University of Rio Grande do Norte, Natal 59078 970, RN, Brazil; 6Department of Microbiology and Parasitology, Federal University of Rio Grande do Norte, Natal 59078 970, RN, Brazil; guedespmm@gmail.com; 7Electron Microscopy, Cell and Chemical Biology, Leiden University Medical Center, 2300 RC Leiden, The Netherlands; r.i.koning@lumc.nl; 8Postgraduate Program in Functional and Structural Biology, Department of Morphology, Federal University of Rio Grande do Norte (UFRN), Natal 59078 970, RN, Brazil

**Keywords:** extracellular vesicles (EVs), M2-TAM, M1-macrophage, doxorubicin, hyaluronic acid, carvedilol, metastasis, breast cancer

## Abstract

One of the main reasons for cancer’s low clinical response to chemotherapeutics is the highly immunosuppressive tumor microenvironment (TME). Tumor-ass ociated M2 macrophages (M2-TAMs) orchestrate the immunosuppression, which favors tumor progression. Extracellular vesicles (EVs) have shown great potential for targeted therapies as, depending on their biological origin, they can present different therapeutic properties, such as enhanced accumulation in the target tissue or modulation of the immune system. In the current study, EVs were isolated from M1-macrophages (M1-EVs) pre-treated with hyaluronic acid (HA) and the β-blocker carvedilol (CV). The resulting modulated-M1 EVs (MM1-EVs) were further loaded with doxorubicin (MM1-DOX) to assess their effect in a mouse model of metastatic tumor growth. The cell death and cell migration profile were evaluated in vitro in 4T1 cells. The polarization of the RAW 264.7 murine macrophage cell line was also analyzed to evaluate the effects on the TME. Tumors were investigated by qRT-PCR and immunohistochemistry. MM1-DOX reduced the primary tumor size and metastases. NF-κB was the major gene downregulated by MM1-DOX. Furthermore, MM1-DOX reduced the expression of M2-TAM (CD-163) in tumors, which resulted in increased apoptosis (FADD) as well as decreased expression of MMP-2 and TGF-β. These results suggest a direct effect in tumors and an upregulation in the TME immunomodulation, which corroborate with our in vitro data that showed increased apoptosis, modulation of macrophage polarization, and reduced cell migration after treatment with M1-EVs combined with HA and CV. Our results indicate that the M1-EVs enhanced the antitumor effects of DOX, especially if combined with HA and CV in an animal model of metastatic cancer.

## 1. Introduction

Due to its high incidence and mortality rates, cancer presents a persistent health problem around the world. Worldwide, an estimated 19.3 million new cancer cases and almost 10.0 million cancer deaths occurred in 2020. In these estimations, female breast cancer was the most commonly diagnosed cancer, with an estimated 2.3 million new cases (11.7%) [1].

The currently available technologies for early diagnosis and the variety of therapeutic interventions have improved the outcome of the disease, but the search for new products and therapeutic alternatives still continues. Immunotherapy, in particular, has been one of the most successful approaches to treating cancer by the use of different strategies to enhance the immune response against tumors [2,3]. In line with this, the tumor microenvironment (TME) has been studied in depth to understand the dynamics and contribution of the different infiltrating cell populations in order to improve targeted therapies, including immunotherapy. Previous studies have shown that infiltrating immune cells, particularly macrophages, play a role in the carcinogenic process with either anti or pro-tumoral activities [4,5]. Depending on phenotype and surface marker expression, macrophages can be generally classified as M1 or M2 [5]. M1 macrophages display important antitumor activities and are involved in the immune system’s response to cancer, whereas M2 macrophages (also known as tumor-associated macrophages or M2-TAMs) promote cancer progression and contribute to multidrug resistance in cancer therapy [5,6]. Therefore, multiple antitumoral approaches are being explored to induce the shift of M2-TAMs to proinflammatory M1 macrophages [7,8,9]. Among the new immunotherapeutic treatments, approaches based on extracellular vesicles (EVs) are receiving attention for their antitumor potential and the bioengineering possibilities that they can offer [10,11,12]. EVs have been efficiently used as carriers and tissue-specific drug delivery systems for antitumor agents [13,14]. In the same way, EVs derived from different immune cells as well as directly from tumor cells are able to activate the immune system by exposing tumor antigens or immune-enhancing molecules and can further be engineered to potentiate or redirect its antitumor effects, an interesting concept known as “tumor vaccines” [12,15,16]. In addition to their biomedical advantages and immunomodulatory properties, some EVs have also demonstrated intrinsic antitumor activity due to specific molecules and tumor suppressor agents that they contain, which are derived from their parental cell [12,13,17].

EVs derived from M1 macrophages have demonstrated important antitumor properties against different types of cancer, including melanoma and breast cancer [18]. M1-derived EVs (M1-EVs) repolarize M2 macrophages and activate the resident M1 macrophages, thereby creating a proinflammatory TME that decreases tumor growth and induces apoptosis in cancer cells [7,8]. M1-EVs are also efficient adjuvants and nanocarriers that enhance the anticancer properties of cytostatic drugs and monoclonal antibodies in vitro and in vivo [8,19]. Evidence has suggested that hyaluronic acid (HA) can be used to increase targeting through the CD44 receptor and promote antitumor macrophage polarization [20,21,22]. On the other hand, carvedilol (CV) is a β-blocker or a β-adrenergic receptor (β-AR) antagonist and is traditionally used for cardiovascular disorders [23]. The polarization of macrophages to an M2 phenotype is mediated by β-adrenergic receptors [24], which means that CV is a promising compound to downregulate TAMs via β-adrenergic receptors in the TME.

Based on this evidence, in the present work, we explored the in vitro and in vivo antitumor and antimetastatic properties of EVs from classically activated M1 macrophages, alone and loaded with doxorubicin (DOX), in breast cancer models. We also evaluated how these anticancer effects can be potentiated by the chemical modulation of the M1 polarization using therapeutic compounds, such as HA and CV.

## 2. Materials and Methods

### 2.1. Antibodies and Reagents

Mouse anti-E-cadherin (Cat # MA1-06304) and mouse anti-vimentin (Cat # PA5-96191) were purchased from Thermo Fisher Scientific (Waltham, MA, USA). Anti-mouse CD163- PerCP (Cat # 46-1631-82) and anti-mouse CD68-FITC (Cat # 11-0689-42) were obtained from eBioscience (San Diego, CA, USA). Recombinant mouse IL-4 (Cat # 214-14) and IFN-**γ** (Cat # 315-05) were purchased from PeproTech (Rocky Hill, NJ, USA). Mitomycin (Cat # M5353) was purchased from Sigma-Aldrich (Amsterdam, The Netherlands).

For immunofluorescence labeling, purified rat anti-E-cadherin (Cat # 13-5700), the secondary antibodies goat anti-mouse (Cat # A32723), and goat anti-rat Alexa^®^ 488 (Catalog # A-11006, Thermo Fisher Scientific) were used. Rabbit anti-β-actin (Cat # 926-42210, LI-COR Biosciences, Lincoln, NE, USA), GRP94 (Cat # MA3-016, ThermoFisher), HSP70 (Cat # 4872S, Cell Signaling, Danvers, MA, USA), Flotillin-1 (Cat # 3244S, Cell Signaling), Integrin α6 (Cat # 3750S, Cell Signaling) and Integrin β1 (Cat # 4706S, Cell Signaling) were used for immunoblotting.

### 2.2. Cell Lines and Cell Culture

Murine breast cancer cells (4T1) and the mouse macrophage RAW 264.7 cell line were obtained from ATCC (Rockville, MD, USA). Cells were grown in an incubator and maintained at 37 °C, and 5% CO_2_ with Dulbecco’s modified Eagle’s medium (DMEM, Gibco Laboratories, Grand Island, NY, USA) supplemented with 1% penicillin/streptomycin and 10% fetal bovine serum (FBS) (Gibco Laboratories, Grand Island, NY, USA). Cells were passaged biweekly using trypsin/EDTA in phosphate-buffered saline (PBS).

### 2.3. Preparation of Single and Modulated EVs and DOX Loading

RAW 264.7 cells were grown to a density of 18.4 × 10^6^ cells in 225 cm^2^ flasks with DMEM (1x) + GlutaMAX (Gibco) supplemented with 10% FBS and 1% penicillin/streptomycin at 37 °C and 5% CO_2_. To obtain EVs from M1 or M2 macrophages, cells were incubated with either 0.04 µg/mL IL-4 for M2-derived EVs (Peprotech, lot# 021749 J2418) or a combination of LPS and 0.1 µg/mL IFN-γ for M1-EVs for 48 h in 25 mL serum-free DMEM. Cells were washed and incubated for an additional 48 h with either serum-free media (for M1 and M2 EVs), 100 μg/mL HA, 75 μg/mL CV, or a combination of both (HA + CV). The resulting medium was centrifuged at 2000× *g* at 4 °C for 20 min, followed by 4000× *g* at 4 °C for 20 min. Afterward, the pellet was discarded to remove cell debris. The medium was then incubated with an EV precipitation reagent overnight (Exo-spin^TM^ Buffer, Cell Guidance Systems, cat# EX06-250). EVs were precipitated by centrifugation at 16,000× *g* for 60 min, resuspended in 100 μL of PBS, and purified using size exclusion Exo-spin columns (Cellgs^®^) according to the manufacturer’s protocol.

To incorporate DOX, freshly isolated EVs (M1 and MM1) were incubated with an 8 mM solution of DOX (Actavis) in 100% Dimethyl Sulfoxide (DMSO, SigmaAldrich, cat# D2650-100ML), in a 2:8 ratio, at 4 °C for 1 h. Afterward, EVs were centrifuged using size exclusion columns as described above and stored at −80 °C until used.

### 2.4. Characterization of EVs

Samples were analyzed with a Nanosight^®^ NS300 (Malvern, Almelo, The Netherlands) to evaluate size distribution and concentration by means of particles/mL. For the measurements, samples were first diluted 200-fold in PBS, loaded into an automatic syringe pump, and measured using a camera level of 9 and a detection threshold of 3. Size distribution was analyzed by dynamic light scattering (DLS) using a Zetasizer Nano-ZS (Malvern, Almelo, The Netherlands). To this end, samples were diluted as described above, loaded in the Zetasizer using a disposable polycarbonate capillary cell (DTS 1061, Malvern, Almelo, The Netherlands), and measured using automatic settings. To analyze morphology, samples were also analyzed using cryo-EM using a Tecnai 12 electron microscope (FEI Company, Eindhoven, The Netherlands) as previously described [11]. All images were recorded at 18,000× magnification (pixel size 1.2 nm) operating the microscope at 120 kV. The concentration of DOX in EV samples was measured using a UV-visible spectrophotometer (SpectraMax ID3 microplate reader, Molecular Devices, San Jose, CA, USA). Free DOX was dissolved in DMSO in serial dilutions, while EV-DOX samples were lysed and dissolved in DMSO. All samples were measured in the microplate reader at 480 nm, and a calibration curve was used to extrapolate the DOX concentration. To evaluate total protein content, samples were incubated with a mixture of 20 mM HEPES, 0.5 mM phenylmethylsulfonyl fluoride (PMSF), and 50 mM phosphatase inhibitor ortho-vanadate dissolved in PBS. Samples were sonicated for 5 min, centrifuged at 16,000× *g* for 5 min, and the supernatant was used for analysis using a MicroBCA protein assay kit (Thermo Fisher) according to the manufacturer’s instructions. Protein characterization was performed using capillary electrophoresis as previously described [10,11] using Wes^®^ automated Western Blot Testing (ProteinSimple, San Jose, CA, USA). Briefly, 0.8 µg/µL of proteins were prepared with a fluorescent master mix containing SDS/DTT and loaded into a provided microwell plate. Primary antibodies (rabbit); anti-Flotillin-1 (Cell signal, 1:10), anti-HSP70 (Cell signal, 1:100), anti-α6-integrin (Cell signal 1:10), and anti-β-actin (1:50 BioLegend, San Diego, CA, USA) were diluted in the supplied antibody diluent and loaded into the plate. Subsequently, the provided secondary antibodies, luminol/peroxidase blocking buffer, and HRP streptavidin (anti-rabbit detection kit, protein simple) were loaded and measured according to the manufacturer’s instructions using 25-capillary cartridges (SM-W004). Quantification of the bands was performed using the compass software (protein simple) by measuring the area of the chemiluminescent bands and calculating the relative expression to control samples.

### 2.5. Polarization of RAW 264.7 Cells

RAW 264.7 cells were grown in a 12-well plate at a density of 5 × 10^5^ cells and supplemented with DMEM as described above for 24 h. For induction of M2-like polarization, cells were cultured with 0.04 µg/mL of IL-4 for 48 h in serum-free medium. The supernatant was collected to be used as M2-conditioned medium (CM) for further studies. For M1-like polarization, RAW 264.7 cells were incubated with 0.1 μg/mL lipopolysaccharides (LPS) and 0.1 μg/mL interferon gamma (IFN-γ) for 48 h.

To evaluate the effect of EVs in macrophage polarization, RAW 264.7 cells were also incubated with RAW-EVs, M1-EVs, M1-HA-EVs, M1-CV-EVs, and M1-HA-CV-EVs (MM1-EVs) for 48 h. After that time, all cells were collected, washed in PBS, and resuspended in buffer containing PBS with 0.5% bovine serum albumin (BSA) and 0.02% sodium azide. Finally, cells were stained using anti-mouse CD163-PerCP, anti-mouse (M2 polarization), and CD68-FITC (for M1 polarization) and analyzed by means of flow cytometry on an LSR II (BD Biosciences, San Jose, CA, USA).

### 2.6. Cell Viability Assays

4T1 and RAW 264.7 cells were seeded at 3 × 10^3^ cells/well in 96-well plates 48 h prior to the viability assays. 4T1 cells were incubated for another 48 h with either M1, M2, or RAW-EVs, as well as EVs from M1 macrophages modulated with the therapeutics compounds HA, CV, or both (MM1-EVs). Samples were also evaluated after loading with 20 µg/mL of DOX. EV samples (without DOX) were added at serial dilutions according to their protein concentration (20 µg/mL; 10 µg/mL; 5 µg/mL), while samples with DOX were added according to the DOX concentration (15 µM; 7.5 µM and 3.75 µM). On the other hand, RAW cells were also treated with 100 μg/mL HA, 75 μg/mL CV, or a combination of both (HA + CV) for 48 h. After the incubation period, the medium was refreshed for both cell lines, and cells were incubated with CellTiter 96 AQueous One Solution (MTS) solution (Promega, Madison, WI, USA) according to the manufacturer’s guidelines. Absorbance was measured at 490 nm using a (SpectraMax ID3 microplate reader, Molecular Devices).

### 2.7. Wound-Healing Assay

To evaluate the effect of EVs on migration, 4T1 cells were seeded in a 12-well plate at a density of 5 × 10^5^ cells and supplemented with DMEM for 24 h. Each well was scratched vertically with a 20–200 µL pipette tip to produce a wound and incubated with 10 μg/mL mitomycin C for 2 h (except for the control cells). Cells were then washed with PBS and incubated with 20 µg/mL of either RAW, M1, or M2 cell-derived EVs for an additional 24 h. Cells were imaged with an Olympus IX70 light microscope (Olympus, Shinjuku, Japan), and the number of cells that migrated into the wound area was calculated using the ImageJ software (https://imagej.nih.gov/ij/ accessed on 10 February 2022) as previously described [5].

### 2.8. Immunofluorescence

To address the effect of EVs on vimentin and E-cadherin expression, 4T1 cells were seeded on glass coverslips in 12-well plates at a density of 5 × 10^5^ cells. After 24 h, cells were incubated with 20 µg/mL of either RAW, M1, or M2-EVs in serum-free DMEM/CM at a ratio of 1:1 for an additional 48 h. To prepare cells for imaging, samples were fixated with 1% paraformaldehyde in PBS, incubated with PBS tween-20 (0.05%, Sigma-Aldrich, Saint Louis, MO, USA), and permeabilized by incubating with 0.1% Triton X-100 (Sigma-Aldrich) for 10 min. Samples were then incubated overnight with primary anti-mouse E-cadherin (1:100) in blocking solution containing 5% normal goat serum (Dako, Glostrup, Denmark) and 0.1% Triton X-100. The next day, samples were washed in blocking solution for 10 min and incubated with goat anti-mouse Alexa^®^ Fluor 488 at 1:300 in blocking solution for another 60 min and 1:1000 DAPI (Life Technologies, Paisley, UK) for nuclear staining. Samples were analyzed using a Leica DM5500 B fluorescence microscope (Leica Microsystems, Wetzlar, Germany) using an A filter cube (Leica Microsystems), allowing excitation at 350 nm for DAPI staining and an L5 filter cube (Leica Microsystems) allowing excitation at 488 nm for E-cadherin.

### 2.9. In Vivo Study

#### 2.9.1. Animals

For orthotopic breast cancer induction studies, 7–9-week-old female BALB/c mice weighing between 21 and 28 g were purchased from the Keizo Asami Immunology Laboratory Biotery (FIOCRUZ-PE) and used for orthotopic breast cancer induction studies. Animals were housed in cages with free access to food and water and treated according to the ethical principles for animal experimentation. All surgical and experimental procedures were approved by the Committee on the Ethics of Animal Experiments of the Federal University of Rio Grande do Norte ethics committee (CEUA, permit number: 11/05/2020).

#### 2.9.2. Orthotopic Tumor Induction

To induce the orthotopic breast cancer model, 35 female BALB/c mice aged 7–9 weeks and weighing 21–28 g were inoculated with murine 4T1 cells. Initially, the cells were cultured in flasks containing DMEM supplemented with 10% FBS and collected at about 70% confluence. Using a syringe, 4T1 cells (1 × 10^6^ cells/100 µL) were inoculated slightly below the fourth left breast of anesthetized animals (xylazine and ketamine) [6]. Tumor growth was monitored daily with a caliper, and upon reaching 3 mm in diameter, animals were divided into seven groups and treated peritumorally with (01) sterile saline, (02) 5 mg/kg DOX, (03) 2 mg/kg M1-DOX, and (04) 2 mg/Kg MM1-DOX. The treatments were readministered every 5 days (3 treatments). During treatment, the diameter of the tumors was measured every two days. Five days after the last treatment (day 21), the animals were euthanized, and then blood was collected from the cardiac cavity for biochemical analysis. In addition, the tumors, lungs, and livers were harvested for TME analysis by means of qRT-PCR and immunohistochemistry or immunofluorescence. Results were expressed as a growth curve from the average tumor volume (mm^3^) calculated by the following equation according to volume mm^3^ = (width × length^2^) × 0.52 [6]. Metastatic niches in the liver and lung were assessed semi-quantitatively as previously described [6,25]. Briefly, the percentage of cancer cells in the tissue parenchyma was represented by scores (1, <5%; 2, 5% to 25%; 3, 26% to 50%; 4, 51% to 75%; 5, >75%). The scores were applied to histological images captured in 20 random fields. Histopathological analysis of tumor tissue, liver, and lungs was independently assessed by two non-operator analysts (RA and RC). Three histological sections per animal tissue (*n* = 5) were analyzed in each group.

#### 2.9.3. qRT-PCR

Gene expression analysis was performed on tumors obtained from BALB/c mice with breast cancer. Fragments of the tumors, which were stored in Trizol^®^ reagent (Invitrogen, CA, USA) at −80 °C, were thawed and ground with a drill. The following steps are common for both sample types, which were processed into chloroform and absolute ethanol for complete RNA extraction [6]. Then, total RNA was purified using SV Total RNA Isolation System (Promega, WI, USA) according to the manufacturer’s instructions. Next, the RNA was immediately converted to cDNA by reverse transcriptase using a High-Capacity RNA-to-cDNA™ kit (Applied Biosystems, Waltham, MA, USA). Real-time quantitative PCR analyses of CD8, NFκB, FAAD, and β-actin mRNAs were performed with SYBR-Green Mix in the Applied Biosystems^®^ 7500 FAST system (Applied Biosystems, Foster City, CA, USA). The experiments were performed in triplicate using the following primers: β-actin (forward, 5′-CCACCATGTACCCAGGCATT-3′ and reverse, 5′- CGGACTCATCGTACTCCTGC-3′, annealing primer temperature, 60 °C); CD8 (forward, 5′-GCTCAGTCATCAGCAACTCG-3′ and reverse, 5′-ATCACAGGCGAAGTCCAATC-3′, annealing primer temperature, 59 °C); NFκB (forward, 5′-CCGTCTGTCTGCTCTCTCT-3′, and reverse, 5′-CGTAGGGATCATCGTCTGCC-3′, annealing primer temperature, 60 °C) and FAAD (forward, 5′-AGAAGAAGAACGCCTCGGTG-3′ and reverse, 5′- GCTCACAGATTCCTGGGCTT-3′, annealing primer temperature, 60 °C). The relative expression was calculated using the ΔΔCt formula as previously described [6].

#### 2.9.4. Immunohistochemistry

Three paraffin-embedded fragments of the tumors from five mice per group were cut into 3 µm sections and mounted on glass slides [6]. Tissues were subjected to deparaffinization and rehydration steps followed by incubation in sodium citrate solution (11 µM) at 90 °C for 30 min for antigen retrieval. Endogenous peroxidase inactivation was performed using 3% hydrogen peroxide and subsequent blocking of nonspecific markings with Novocastra Protein Block (Leica Biosystems, Wetzlar, Germany). Tissue sections were incubated with anti-NFKB, anti-MMP-2, anti-CD163, anti-CXCL12, and anti-TGF-β primary antibodies diluted in Diamond antibody diluent (1:400; Cell Marque, Rocklin, CA, USA) overnight at 4 °C. Slides were washed with PBS and incubated with biotinylated pan-specific universal secondary antibody—R.T.U. Vectasian Kit (Vector, Burlingame, CA, USA), followed by streptavidin/HRP-conjugated incubation. Diaminobenzidine (DAB; DAKO, Santa Clara, CA, USA) was used as chromogen. Sections were counter-stained with hematoxylin and imaged using a Nikon E200 LED light microscope (Minato, Tokyo, Japan) coupled to a digital camera (Moticam, Kowloon Bay, Hong Kong), where digital images were captured. The immunoreactivity classification was calculated as previously described [6].

#### 2.9.5. Immunofluorescence Microscopy

The preparation of samples for immunofluorescence was performed as previously described [26]. Briefly, tissue sections from 5 animals per group were deparaffinized, hydrated, and treated with antigen retrieval solution (10 mM sodium citrate with 0.05% Tween 20) at 95 °C for 40 min. Slices were allowed to react with rabbit 1:400 anti-TGF-β (Abcam, Waltham, MA, USA) overnight and subsequently washed 3 times with 0.2% triton X-100 in PBS. Tissues were then incubated with secondary antibody 1:500 Alexa Fluor 488 (goat anti-rabbit) and DAPI 1:1000 DAPI for nuclear staining. Samples were mounted with VECTASHIELD (Vectorlabs, Burlingame, CA, USA) and imaged under a fluorescence microscope (ZEISS, Jena, Germany).

#### 2.9.6. Statistical Analysis

All experiments were performed in triplicate and expressed as mean ± SEM. Data were analyzed using Prism 8.0 (GraphPad Software Inc., San Diego, CA, USA) and calculating the analysis of variance (ANOVA) corrected by Bonferroni’s for multiple testing unless indicated otherwise in the figure legend.

## 3. Results and Discussion

### 3.1. EVs from Polarized Macrophages

EVs isolated from RAW 264.7, as well as M1- and M2-polarized macrophages, were characterized according to size, concentration, and protein expression (Figure 1). The three types of vesicles presented a well-defined round morphology, with a size range of 100–200 nm, and were obtained in a concentration of around 4 to 6 × 10^8^ particles per mL, corresponding to 1–2 mg/mL. Western blot analysis of the EVs revealed the presence of typical EV markers, such as the transmembrane protein flotillin, heat shock protein 70 (HSP70), the intracellular cytoskeleton-associated protein actin, and the adhesion protein integrin α_6_ (Figure 1B). These data are in accordance with our previous observations [10,11] and reports from other authors indicating that our prepared samples were indeed enriched with EVs.

### 3.2. M1 but Not M2-Derived EVs Have Antitumor Potential in Breast Cancer Cells

EVs isolated from RAW 264.7, M1- and M2-polarized macrophages were evaluated according to their antitumor properties in 4T1 cells (murine breast carcinoma). The M1-derived EVs significantly decreased the viability and proliferation of 4T1 cells in a concentration-dependent manner in comparison to M2 or RAW-derived EVs (Figure 2A). The capacity to induce apoptosis was also evaluated by a double staining Annexin/DAPI assay and Western blot analysis of caspase 3 expression. We observed that only EVs from M1 macrophages increased Annexin and Annexin/DAPI staining in 4T1 cells in a significant manner when compared to EVs from M2 macrophages and RAW 264.7 cells (Figure 2B). In accordance with this, M1-derived EVs also decreased the levels of full-length caspase 3, which is an indication of caspase activation and induction of apoptosis (Figure 2C). Thus, our data suggest that EVs isolated from M1 but not M2 macrophages or RAW cells have a natural apoptotic effect on 4T1 cells. The capacity of M1-EVs to promote apoptosis via the caspase 3 pathway was previously shown by Wang et al., 2020 [19], although they did not observe a significant effect on cell viability as the one observed in our study. The enhanced effect observed in Figure 2A,B might be related to a different polarization method (IFN-γ vs. IFN-γ + LPS) and different concentrations of EV used. We also evaluated E-cadherin expression in 4T1 cells in response to the three types of EVs by immunofluorescence. We observed a slight increase in the expression of E-cadherin in M1-EVs when compared to M2 and RAW cell-derived EVs (Figure 2D). The mentioned effect was only surpassed by that of mitomycin, which was used as a positive control to verify the increase in E-cadherin expression. In order to explore this mechanism further, we performed a wound-healing assay using mitomycin as an internal control [5]. As shown in Figure 2E,F, M1-EVs but not EVs derived from RAW or M2 macrophages prevented wound closure to a similar extent to the control + mitomycin. This result is important considering that 4T1 is a highly metastatic cell line, and the migration capacity of cancer cells in vitro is closely related to its metastatic potential in vivo. Interestingly, M2-EVs had the opposite effect, increasing the migration of 4T1 cells, which supports the hypothesis that M2-EVs have pro-tumoral/regenerative properties. Another explanation of the effects of M1-EVs on migration could be that these vesicles can deliver miRNA-326, which is involved in the suppression of migration and invasion through downregulation of the NF-κB expression [27].

### 3.3. Enhanced Antitumor Properties of M1-Derived EVs by Combination of HA and CV

Recent studies suggested that the effect of macrophage polarization can be modulated with multiple drugs, leading to increased efficacy. We hypothesized that, by incubating macrophages with such drugs, we could also produce EVs with enhanced therapeutic potential. For that purpose, we selected two compounds: HA and CV, based on previous reports indicating a modulatory effect on macrophage polarization [6,24]. As shown in Figure 3A, during the polarization process, HA and CV alone or in combination did not induce significant changes in cell viability, indicating that these compounds did not induce cell death in RAW 264.7 cells. EVs from M1 macrophages polarized in combination with HA and/or CV were isolated and subsequently incubated with RAW cells to evaluate their capacity to modulate macrophage polarization. As expected, the incubation of RAW 264.7 cells with LPS + INFγ (used a positive control) increased the percentage of CD68+ cells while treatment with IL4 strongly induced the expression of CD163, indicating the polarization to either an M1 or M2 phenotype, respectively. Interestingly, incubation with EVs from M1 macrophages with or without pre-treatment with HA and CV was also able to induce M1 polarization, and this effect was enhanced using the combination of both compounds simultaneously. Based on these results, MM1-EVs showed higher efficiency in stimulating polarization toward an M1 phenotype than M1-EVs, M1-EVs + HA, or M1-EVs + CV. Therefore, all of the following experiments were performed with MM1-EVs (Figure 3B). Next, we investigated whether these EVs could also downregulate markers of epithelial-mesenchymal transition (EMT) in 4T1 cells. We observed that MM1-EVs upregulated E-cadherin and decreased vimentin expression when compared to M2, RAW, and M1-EVs. As increased vimentin levels and decreased E-cadherin levels are crucial in the EMT process, our data indicate an interesting antimetastatic effect [5,6] (Figure 3C). In the same way, MM1 EVs significantly reduced the viability of 4T1 cells compared to M1 or RAW-derived EVs, which supports the hypothesis that the pre-modulation of macrophages with the combination of HA and CV enhances the antitumoral properties of M1-EVs (Figure 3D). In order to enhance the therapeutic potential even further, EVs from M1 and MM1 macrophages were loaded with DOX, a reference chemotherapeutic drug used for the treatment of soft-tissue cancer, including breast cancer. The physicochemical properties of DOX make this compound ideal for EV encapsulation by direct incubation, which is a simple method that does not require any synthetic steps and has previously been tested in other studies [28,29]. Both M1 and MM1-EVs loaded with DOX (M1-DOX and MM1-DOX) strongly decreased the cell viability of 4T1 cells at lower concentrations and significantly enhanced the effect of DOX without additional compounds. In accordance with our previous observations, MM1-DOX was significantly stronger than M1-DOX (Figure 3E). By using an Annexin V/DAPI double staining, we observed that the combination of EVs with DOX induced a strong apoptotic effect in 4T1 cells, and, particularly, the combination of MM1-DOX was able to produce over 99% of cell death in 4T1 cells (Figure 3F).

### 3.4. The Combination of HA and CV Potentiates In Vivo Anticancer Activity of M1-Derived EVs

To explore the potential of MM1-EVs in combination with DOX in vivo, we generated an orthotopic mouse model of breast cancer by implanting 4T1 cells in the fourth left breast of BALB/c mice. M1-DOX and MM1-DOX were used at a concentration of 2 mg/kg and compared with free DOX at a dose of 5 mg/kg, which had been previously established as an effective dose to reduce tumor growth in this breast cancer model [30]. The experiment revealed that all samples were able to reduce tumor growth when compared to the groups treated with saline solution. In addition, as previously observed in vitro, the antitumor effect of MM1-DOX was significantly higher than M1-DOX and free DOX (Figure 4A,B). These data were corroborated by weighting the isolated tumors ex vivo (Figure 4C).

It has been demonstrated that M1-EVs induce proinflammatory conditions by inducing the release of proinflammatory cytokines (including TNF-α, IL-6, IL-12) and downregulating the expression of M2-TAMs (CD163) in the TME as well as stimulating apoptosis in tumor cells [31,32]. These are three important conditions that contribute to an overall strong anticancer effect. NF-κB is an inhibitor of programmed cell death [33] via transforming growth factor β (TGF-β) [34]. In cancer, elevated NF-κB activity is associated with the downregulation of pro-apoptotic genes such as Fas-associated protein death domain (FAAD) and caspase 8 [35]. Immunologically, the NF-κB signaling pathway plays an important role in the activation of T cells as well as the survival and/or function in the context of cancer cells [36]. The tumor growth may be related to the blocking of activation of CD8 lymphocytes by the inhibition of NF-κB via TGF-β, which was released by M2-TAMs [6,36,37]. Next, we examined the expression of different inflammation, immune, and cell death-related markers in tumor sections from control and treated animals. Our data indicate that MM1-DOX was able to reduce NF-κB expression, as measured by immunohistochemistry (Figure 4D,E) and mRNA expression (Figure 4F), with significant differences in comparison to M1-DOX, free DOX, and saline groups. Interestingly, free DOX caused a slight increase in NF-κB mRNA expression in tumor samples, which, although it was not statistically significant, could be related to the high dose of DOX used for this group (5 mg/kg compared to 2 mg/kg in M1 and MM1). In a similar manner, MM1-DOX increased the mRNA expression of CD8 (Figure 4G) and FADD (Figure 4H), indicating an improvement in the infiltration of cytotoxic lymphocytes and induction of apoptosis in the tumors. Some studies have shown that CV exhibits antitumor effects by inducing cell apoptosis in some cancer cells in vitro as well as downregulating the migration and invasion of breast cancer cells [38,39,40]. On the other hand, HA is a hydrophilic mucopolysaccharide that can be found in the stroma of animal tissues [41]. The CD44 receptor is the main receptor for HA, which is expressed in several malignant cells [42,43]. CD44 receptors are also involved in the development of drug resistance and inhibition of apoptosis due to their effects on the NF-κB signaling pathway [44,45]. MM1-EVs carrying HA, natural proinflammatory cytokines, and CV bind easily to the cancer cell surface, thereby promoting cellular uptake and a potentiated antitumor effect of DOX on the TME. In our study, the groups treated with MM1-DOX showed a significantly higher antitumor effect when compared to groups treated with saline solution, M1-DOX, or free DOX.

### 3.5. Combined Treatment with HA and CV Increases Antimetastatic Capacity of M1-EVs

Our previous observations suggested that EVs derived from polarized macrophages had an antimetastatic potential in vitro. Considering that the orthotopic 4T1 mouse model develops spontaneous metastases and resembles advanced stage breast cancer, we decided to evaluate the antimetastatic properties of our EV preparations in vivo. We used samples from tumors, liver, and lungs to compare different indicators of metastases in response to the different treatments. We observed that the administration of M1-DOX and MM1-DOX significantly decreased the formation of metastatic niches in the liver (Figure 5A,B) and lungs (Figure 5C,D) when compared to groups that only received a high dose of DOX or saline solution. As with our previous observations, the effect of MM1-DOX was statistically higher than M1-DOX.

Furthermore, we investigated the expression of CXCL12 in the lungs and liver, where metastatic niches were observed. CXCR4 and its ligand CXCL12 are strongly involved in the proliferation, survival, and invasion of cancer cells [46]. This axis CXCR4-CXCL12 has been related to the “seed-soil” theory, which explains its role in regulating metastasis of breast cancer to specific organs [47]. We observed that, after the treatment with M1-DOX and mainly MM1-DOX, the expression of CXCL12 significantly decreased in the liver (Figure 5E,F) and lungs (Figure 5G,H) when compared to groups that only received a high dose of DOX or saline solution.

In a similar manner, MM1-DOX significantly decreased the expression of matrix metalloproteinase (MMP)-2 in tumors, as measured by immunohistochemistry, compared to groups treated with saline solution, free DOX and M1-DOX groups (Figure 6A,B). Furthermore, we evaluated the expression of TGF-β using immunofluorescence in tumor sections of control and treated animals. MM1-DOX also showed a significantly reduced expression of TGF-β compared to saline solution, M1-DOX, or free DOX alone (Figure 6C,D). Finally, we found that the expression of CD163 reduced after the treatment with MM1-DOX when compared to saline, M1-DOX, and DOX groups (Figure 6E,F). Although various cancer cells produce TGF-β, which promotes the progression of EMT before metastasis, TGF-β is also secreted by TAMs. This suggests that the TGF-β signaling significantly increases the EMT and metastasis by reducing inflammation and diminishing the activity of CD8+ cells [48]. Furthermore, M2-TAMs potentiate metastasis from primary TME by stimulating the TGF-β/MMP-2 signaling axis in cancer cells [49,50,51]. The MMPs constitute a large family of endopeptidases, such as MMP-2 and MMP-9, which are responsible for degrading almost all extracellular matrix components during the invasion process of cancer cells in the TME [52]. High expression of MMP-2 and TGF-β in human tumors has been correlated with poor prognosis [7,53,54]. In this context, downregulating M2-TAMs through M1-EVs in the TME indicates to be a promising strategy for blocking pro-tumor signals avoiding tumor growth and metastasis. Since EVs contain a multitude of bioactive cargos [12,32], it was to be expected that our HA and CV-loaded EVs were able to downregulate the expression of M2-TAM (CD163) in the TME, as was shown in our study. Similar to the in vitro results, MM1-DOX was more effective in downregulating the expression of M2-TAM (CD163) in the TME than free DOX and M1-DOX, which resulted in a reduction in tumor progression as seen by the decreased metastases in the lungs and liver.

## 4. Conclusions

Our present study identifies the potential of EVs isolated from polarized macrophages. We developed and evaluated the effect of EVs obtained from macrophages polarized with different stimuli, including LPS, IFN-γ, HA, CV, and IL4, in preclinical models of breast cancer. Our data suggest that by using a combination of LPS, IFN-γ, HA, and CV, the percentage of macrophages polarized toward an M1 phenotype is enhanced. The resulting MM1-EVs showed an enhanced effect on apoptosis and reduction in cell viability and migration when compared to M1-EVs and non-polarized EVs. MM1-EVs were loaded with DOX to further explore their therapeutic potential against breast cancer cells and tumors and compared to M1-EVs or a high dose of free DOX. All groups were able to decrease cell viability, increase apoptosis and induce inhibition of tumor growth, but MM1-DOX were the most efficient, showing almost total cell death and strong inhibition of tumor growth. M1-DOX and MM1-DOX treatments reduced NF-κB expressions while increasing mRNA expression of CD8 and FADD in the extracted tumors. In both cases, this effect was higher than DOX alone. Furthermore, MM1-EVs but not M1-EVs had a significatively higher reduction in metastatic niches and expression of TGF-Β, MMP-2, and CD163 compared to a high dose of DOX. Therefore, our data suggest that MM1-DOX had an important immunomodulatory effect in the TME by reducing the expression of M2-TAMs (CD163), increasing CD8+ cells, and downregulating NF-κB, which are key steps in the blockade of metastasis. Overall, our study can be considered a novel strategy to increase the outcomes of cancer therapy by using MM1-EVs combined with therapeutic compounds and a chemotherapeutic agent.

## Figures and Tables

**Figure 1 pharmaceutics-14-01068-f001:**
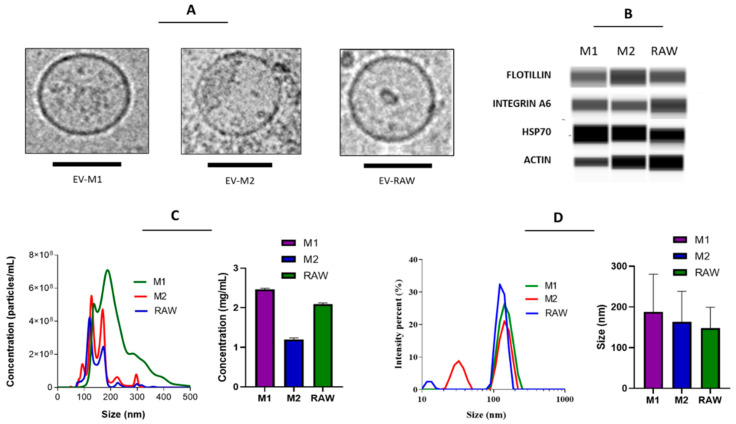
Characterization of EVs derived from RAW 264.7 cells as well as M1- and M2-polarized macrophages. Representative (**A**) cryo-EM micrograph, (**B**) capillary electrophoresis, (**C**) NTA, and (**D**) DLS. Scale bars are 100 nm.

**Figure 2 pharmaceutics-14-01068-f002:**
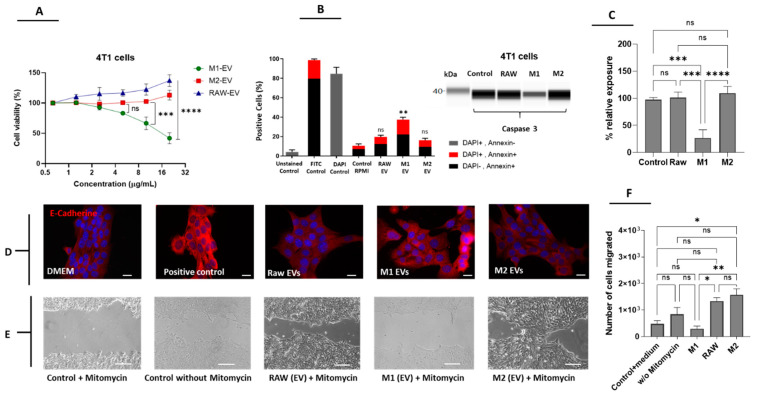
In vitro effects of macrophage-derived EVs on proliferation and migration of 4T1 cells. (**A**) An MTS assay to investigate the effect of M1, M2, and RAW-EVs on cell viability. (**B**) Annexin V-FITC/DAPI double staining was used to evaluate apoptosis induction in response to the EVs. (**C**) Western blot analysis of caspase 3 integrity in cells treated with the three types of EVs (**left**) and relative exposure (**right**). (**D**) Immunofluorescence analysis of E-Cadherin expression in response to EVs (Scale bars are 20 µM). (**E**) Wound-healing assay in the presence of mitomycin to verify the effect of EVs on cell migration (Scale bars are 200 µM). (**F**) Quantification of migrating cells in the wounded area. Data are means ± SEM (*n* = 3). Non-significant (ns); **** *p* < 0.0001; *** *p* < 0.001; ** *p* < 0.01; * *p* < 0.05 (vs. control unless specified).

**Figure 3 pharmaceutics-14-01068-f003:**
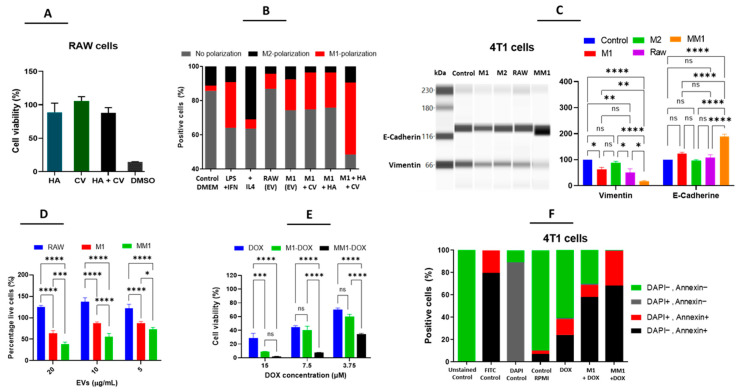
Influence of HA and CV on M1 polarization of macrophages and antitumor potential of EVs in 4T1 cells. (**A**) Cell viability of RAW 264.7 cells treated with HA, CV, or a combination of both. (**B**) Analysis of M1 (by means of CD68 expression) and M2 polarization (by means of CD163 expression) of RAW cells in response to IL-4. LPS + IFNγ, RAW-EVs, and M1-EVs alone and combined with HA, CV, or both. (**C**) Western blot analysis of E-cadherin and vimentin levels in 4T1 cells in response to EVs (**left**) and relative exposure (**right**). (**D**) Cell viability of 4T1 cells in response to RAW, M1, and MM1-EVs. (**E**) Cell viability of 4T1 cells in response to M1 and MM1-EVs loaded with DOX and free DOX. (**F**) Annexin V-FITC/DAPI double staining to evaluate apoptosis induction in 4T1 cells in response to free DOX and DOX-loaded M1 and MM1-EVs. Data are means ± SEM (*n* = 3). Non-significant (ns); **** *p* < 0.0001; *** *p* < 0.001; ** *p* < 0.01; * *p* < 0.05.

**Figure 4 pharmaceutics-14-01068-f004:**
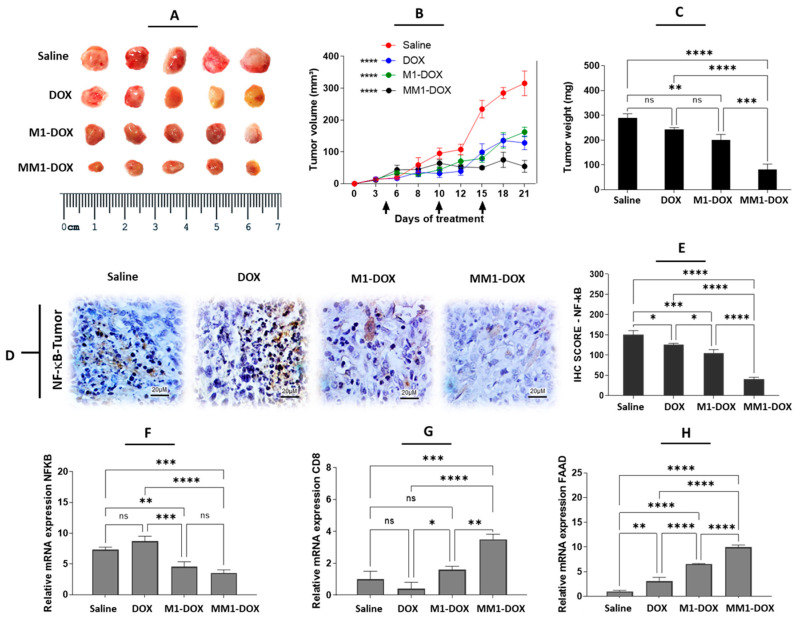
Influence of HA acid and CV on in vivo anticancer properties of M1-EVs loaded with DOX in an orthotopic mouse model of 4T1 pro-metastatic breast cancer. Animals were treated with saline solution (control), high dose of DOX (5 mg/kg), 2 mg/kg of M1-DOX (M1-EVs loaded with 20 µg/mL of DOX), and 2 mg/kg of MM1-DOX (MM1-EVs loaded with 20 µg/mL of DOX). (**A**) Size comparison of the tumors isolated from animals of the four experimental groups. (**B**) Tumor volume of treated animals during the 21 days of the experiment. (**C**) Tumor weight in mg of extracted tumors from animals of the four experimental groups. (**D**) Immunohistochemical (IHC) detection of NF-κB in tumor sections from treated animals. (**E**) IHC score of NF-κB for the four experimental groups. (**F**) Relative mRNA expression of NF-κB, (**G**) CD8, and (**H**) FADD in tumor sections from treated animals. Data are means ± SEM (*n* = 5 animals per condition). Non-significant (ns); **** *p* < 0.0001; *** *p* < 0.001; ** *p* < 0.01; * *p* < 0.05.

**Figure 5 pharmaceutics-14-01068-f005:**
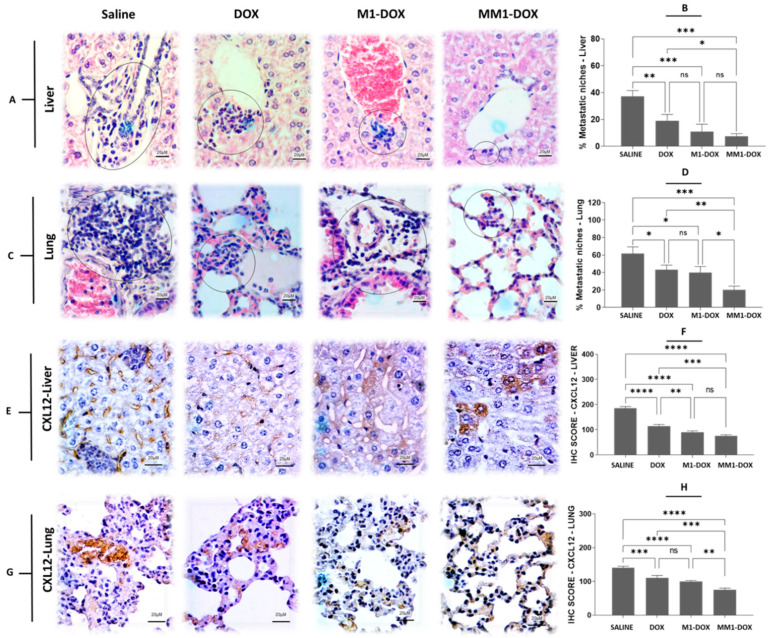
Influence of HA and CV on in vivo antimetastatic properties of M1-derived EVs loaded with DOX in orthotopic 4T1 breast cancer-bearing mice. (**A**) H&E staining of liver sections from treated animals showing the formation of metastatic niches. (**B**) Percentage of liver metastatic niches for the four experimental groups. (**C**) H&E staining of lung sections from treated animals showing the formation of metastatic niches. (**D**) Percentage of lung metastatic niches for the four experimental groups. (**E**) Immunohistochemical (IHC) detection of CXL12 in liver sections from treated animals. (**F**) IHC score for CXL12 from the four experimental groups. (**G**) IHC detection of CXL12 in lung sections from treated animals. (**H**) IHC score for CXL12 from the four experimental groups. Scale bars are 20 µM. Data are means ± SEM. Non-significant (ns); **** *p* < 0.0001; *** *p* < 0.001; ** *p* < 0.01; * *p* < 0.05.

**Figure 6 pharmaceutics-14-01068-f006:**
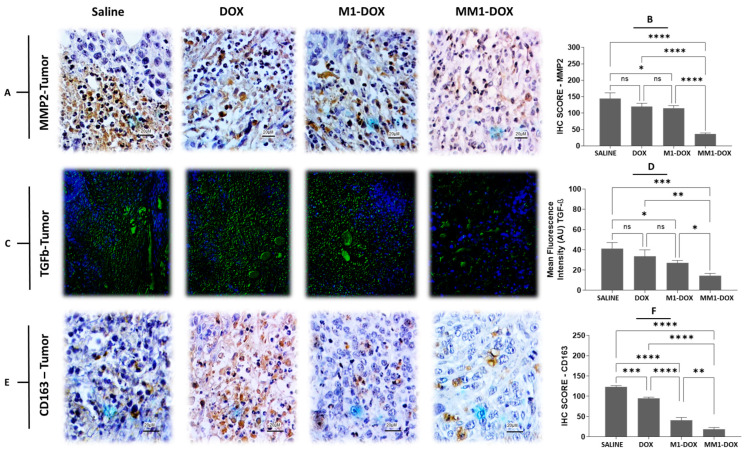
In vivo antimetastatic properties of MM1-derived EVs loaded with DOX in orthotopic 4T1 breast cancer-bearing mice in vivo. (**A**) Immunohistochemical (IHC) detection of MMP-2 in tumor sections from treated animals. (**B**) IHC score for MMP-2 from the four experimental groups. (**C**) Immunofluorescence detection of tumor growth factor beta (TGF-β) in tumor sections of treated animals. (**D**) Mean fluorescence intensity of TGF-β in the four experimental groups. (**E**) IHC detection CD163 in tumor sections from treated animals. (**F**) IHC score for CD163 from the four experimental groups. Scale bars are 20 µM. Data are means ± SEM (*n* = 5 mice per condition). Non-significant (ns); **** *p* < 0.0001; *** *p* < 0.001; ** *p* < 0.01; * *p* < 0.05.

## Data Availability

The raw/processed data required to reproduce these findings cannot be shared at this time as the data also form part of an ongoing study.

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
