# Peer review of "Extracellular Vesicles from M1-Polarized Macrophages Combined with Hyaluronic Acid and a β-Blocker Potentiate Doxorubicin’s Antitumor Activity by Downregulating Tumor-Associated Macrophages in Breast Cancer"

_pharmaceutics, 2022, doi:10.3390/pharmaceutics14051068_

Round 1
Reviewer 1 Report
The manuscript by Jorquera-Cordero et al. investigates the use of extracellular vesicles derived from macrophages as a novel treatment in a model of breast cancer. The study addresses an important issue and most conclusions are supported by the data. The following aspects have to be addressed before publication of the manuscript:
- The rationale for the use of hyaluronic acid and carvedilol needs to be provided in the introduction or in M&M. Currently this information is provided quite late in the manuscript with the presentation of the results.
- Section 2.1. Please, provide the clone or catalogue number of all antibodies used in this study
- Section 2.3, line 119. Please, specify the exact treatment length.
- Line 147. The section on “Characterization of EVs” lacks numbering.
- Line 164. Please, indicate the wavelength used for the determination of doxorubicin concentration.
- Further detail on the determination of the concentration of EVs needs to be provided. Was actually the EV protein concentration measured?
- Line 187, please specify the device used.
- Line 192, a reference is missing
- Lines 361-364. The statement is not clear. According to figure 2, an increase in apoptosis by M1-EVs is observed.
- Line 367. A reference is missing
- Figure 2, please indicate the meaning of the different statistical significance values.
- Figures 22 C and D, please provide histograms with quantitative information of at least 3 independent replicates.
- Section 3.3. It is not clear whether CD63+ or CD163+ cells were studied, please check.
- Section 3.3. The authors state that M1 macrophages with the combination of HA and CV (MM1) showed higher efficiency in blocking M2 polarization. This statement needs to be reformulated. Blocking of M2 polarization was not directly studied.
- Figure 3C, Please, provide histograms with quantitative information of at least 3 independent replicates.
- Figure 3E, Please indicate whether the concentration corresponds to EV or Dox concentration
- Section 3.4. The authors indicate that Dox alone caused an increase in NFkb expression. However, no significant differences are observed in figure 4G, please check.
- Section 3.4. Please check whether the references to the figures are correct. CD8 expression data seems to be presented in figure 4F and not 4G as stated in the section 3.4.
- Figure 5A and B. Further details on the quantification of the metastatic niches need to be provided. It is not clear which situation is considered as “100% of metastatic niches”
Author Response
The rationale for the use of hyaluronic acid and carvedilol needs to be provided in the introduction or in M&M. Currently this information is provided quite late in the manuscript with the presentation of the results.
Thank you for this comments. We have updated the manuscript and properly introduce HA and CV in the introduction section
Section 2.1. Please, provide the clone or catalogue number of all antibodies used in this study
Catalogue numbers are now included in section 2.1
Section 2.3, line 119. Please, specify the exact treatment length.
Treatment length was correctly specified in all the methodology
Line 147. The section on “Characterization of EVs” lacks numbering.
Thank you for the comments. This is now corrected in the manuscript
Line 164. Please, indicate the wavelength used for the determination of doxorubicin concentration.
DOX wavelength was correctly specified in the manuscript as well as additional details about the equipment used for measurements
Further detail on the determination of the concentration of EVs needs to be provided. Was actually the EV protein concentration measured?
Further detail was added as requested. Information about microBCA analysis for protein quantification was missing in the manuscript
Line 187, please specify the device used.
Device used was specified in the manuscript
Line 192, a reference is missing
Missing references was added as requested
Lines 361-364. The statement is not clear. According to figure 2, an increase in apoptosis by M1-EVs is observed.
Statement was corrected in the manuscript as it was referring to non-significant data from other authors.
Line 367. A reference is missing
Missing references was incorporated
Figure 2, please indicate the meaning of the different statistical significance values.
Thank you for this comment. Meaning of statistical significance was added in all figures
Figures 22 C and D, please provide histograms with quantitative information of at least 3 independent replicates.
Histogram with quantitative information (and statistical analysis) related to the capillary electrophoresis and migration assays are now included as requested in figures 2C and 2FSection 3.3. It is not clear whether CD63+ or CD163+ cells were studied, please check.
Both were used. CD63 was used as an indicator of M1-like polarization while CD163 was used for M2. A more detailed explanation is now provided in the manuscript
Section 3.3. The authors state that M1 macrophages with the combination of HA and CV (MM1) showed higher efficiency in blocking M2 polarization. This statement needs to be reformulated. Blocking of M2 polarization was not directly studied.
Thank you for this comments, we corrected the sentence in the manuscript
Figure 3C, Please, provide histograms with quantitative information of at least 3 independent replicates.
Thank you for this comment, we added quantitative information as requested with statistical analysis which was also updated in the methodology.
Figure 3E, Please indicate whether the concentration corresponds to EV or Dox concentration
Figure 3E was updated to indicate that the concentration was refereed to DOX. This was also corrected in the methodology
Section 3.4. The authors indicate that Dox alone caused an increase in NFkb expression. However, no significant differences are observed in figure 4G, please check.
Thank you for this comment. We corrected this sentence and specify in the manuscript that this was a small and not-statistically significant increase which could still be related to the high dose of DOX used.
Section 3.4. Please check whether the references to the figures are correct. CD8 expression data seems to be presented in figure 4F and not 4G as stated in the section 3.4.
Thank you for this comment. We realize that there was a mistake in the figure 4 order. This was fixed and updated in the manuscript
Figure 5A and B. Further details on the quantification of the metastatic niches need to be provided. It is not clear which situation is considered as “100% of metastatic niches”
Dear Reviewer, we are incredibly grateful for these considerations, they contributed greatly to the final result of this work. Your statement about the quantification of the metastatic niches is correct. We performed a semi-quantitative analysis to estimate the approximate amount of metastatic cells or metastatic niches in liver and lungs. To order to understand the effects of our treatment in the metastasis of breast cancer in the BALB/c mice, we evaluated all the extension the tissue parenchyma of these organs and based on percentage of tumour cells scattered on the tissue, we attributed Scores such as: (1, <5%; 2, 5% to 25%; 3, 26% to 50%; 4, 51% to 75%; 5, >75%) as described by Van den Eynden et al., 2012r and Cavalcante et al., 2021p. The scores were applied to histological images captured in 20 random fields. Histopathological analysis of tumor tissue, liver, and lungs were independently assessed by two non‐ operator analysts (RA and RC). Three histological sections per animal tissue (n = 5) were analyzed in each group. As expected, the groups of metastatic cells were preferentially located in the portal space of the liver and in the basal lamina of the pulmonary veins, which demonstrates an early tropism of tumor metastasis in these organs.
With the goal to clarify this statement, we added a text in the 2.9.2 section. “Metastatic niches in the liver and lung were assessed semiquantitatively as described (Van den Eynden et al., 2012 and Cavalcante et al., 2021). Percentage of tumour cells in the tissue parenchyma represented by Scores was applied (1, <5%; 2, 5% to 25%; 3, 26% to 50%; 4, 51% to 75%; 5, >75%). The scores were applied to histological images captured in 20 random fields of all the extension the organs. Histopathological analysis of tumour tissue, liver, and lungs were independently assessed by two non‐operator analysts (RA and RC). Three histological sections per animal tissue (n = 5) were analysed in each group”.R= Van den Eynden, G. G. , Bird, N. C. , Majeed, A. W. , Van Laere, S. , Dirix, L. Y. , & Vermeulen, P. B. (2012). The histological growth pattern of colorectal cancer liver metastases has prognostic value. Clinical & Experimental Metastasis, 29(6), 541–549. 10.1007/s10585-012-9469-1
P= Cavalcante, Rômulo S et al. “STAT3/NF-κB signalling disruption in M2 tumour-associated macrophages is a major target of PLGA nanocarriers/PD-L1 antibody immunomodulatory therapy in breast cancer.” British journal of pharmacology vol. 178,11 (2021): 2284-2304. doi:10.1111/bph.15373

Reviewer 2 Report
It is very interesting paper and easy to read because of its smart structure, etc., but I could not understand only Figure 5. Due to the limitation of the number of characters, please let us know if you can just respond to us privately, even if it is not reflected in the paper.
・What does it mean? "Cells were placed in syringes (1 x 106 cells/100 μl) and inoculated slightly below the fourth left breast". → Is the place specific to induce EMT of 4T1 cells to lung?
・Please describe the concentration of LPS.
・If you add M1-EVs to other cells not cancer cell lines, what happens? Does it induce apoptosis only in cancer cells?
・Please tell me how to find the "metastatic-4T1 cells" in liver or lung. Why do not you show the raw images of liver or lung such as Figure.4A. It surely looks like the niches, but you only studied HE staining. If the primary 4T1 cells stained HERS2 or some cancer-related proteins, why did not you stain the same proteins by IHC? In addition, I am very curious about the data of Figure. 5B and Figure. 5D in saline. The standard deviation was very small. I do not think the metastatic model of 4T1 injection is such the uniform. Again, please tell me how to decide the metastatic niche.
・Then, what is the component of EVs of M1? What is the difference of EVs from M1 vs M2 macrophages? You only showed the phenomena, and I am very interesting, but please discuss or speculate the differences of components. For example, EVs-M1 suppress MMP-2 -> High expression of TIMPs?
Author Response
Reviewer 2
It is very interesting paper and easy to read because of its smart structure, etc., but I could not understand only Figure 5. Due to the limitation of the number of characters, please let us know if you can just respond to us privately, even if it is not reflected in the paper.
・What does it mean? "Cells were placed in syringes (1 x 106 cells/100 μl) and inoculated slightly below the fourth left breast". → Is the place specific to induce EMT of 4T1 cells to lung?
Thank you for your comments. This text is updated for easier understanding in the methodology. The 4T1 orthotopic breast cancer spontaneous metastasis mouse model with the cell line 4T1 highly invasive is such a transplanted tumor model. In this model, the breast tumor cells are transplanted into the mammary fat pad to establish primary tumor nodules a,b.
References
a) Cavalcante RS, Ishikawa U, Silva ES, Silva-Júnior AA, Araújo AA, Cruz LJ, Chan AB, de Araújo Júnior RF. STAT3/NF-κB signalling disruption in M2 tumour-associated macrophages is a major target of PLGA nanocarriers/PD-L1 antibody immunomodulatory therapy in breast cancer. Br J Pharmacol. 2021 Jun;178(11):2284-2304. doi: 10.1111/bph.15373. Epub 2021 Mar 31. PMID: 33434950; PMCID: PMC8251773.
b) Paschall AV, Liu K. An Orthotopic Mouse Model of Spontaneous Breast Cancer Metastasis. J Vis Exp. 2016;(114):54040. Published 2016 Aug 14. doi:10.3791/54040
・Please describe the concentration of LPS.
LPS was used at 0.1 ug/mL. Information is now updated in section 2.3 (methods).
・If you add M1-EVs to other cells not cancer cell lines, what happens? Does it induce apoptosis only in cancer cells?
Although we (and other authors) have observed selectivity in cancer cells we cannot state yet that there is not apoptosis in normal cells. For example we observed no apoptosis in RAW and HEK cells but we did observe a similar effect in the apoptosis of 3T3 cells (data not shown). As showed in this manuscript the effect of M1-EVs seems to be related with the modulation of apoptosis and migration by down-regulating NF-κB expression. In addition to this, M1-EVs are able to induce macrophage polarization towards an M1-phenotye (data non shown in this manuscript). As the TME is highly immunosuppressive, by downregulating NF-κB signaling and stimulate immune cells, the effect on cancer cells will have a high selectivity. Another advantage of using EVs in general is their passive cancer-targeting due to the enhanced permeability and retention (EPR) effect. The small size close to 150 nm allows them to be retained in tumor tissue as they have a high permeability and low lymphatic drainage. This effect is overall known for most of the nanoparticles and confers some additional selectivity over tumors. Please check the following reference for additional information about this topic
Pharmaceutics. 2020 Oct 26;12(11):1022. doi: 10.3390/pharmaceutics12111022.
・Please tell me how to find the "metastatic-4T1 cells" in liver or lung. Why do not you show the raw images of liver or lung such as Figure.4A. It surely looks like the niches, but you only studied HE staining. If the primary 4T1 cells stained HERS2 or some cancer-related proteins, why did not you stain the same proteins by IHC? In addition, I am very curious about the data of Figure. 5B and Figure. 5D in saline. The standard deviation was very small. I do not think the metastatic model of 4T1 injection is such the uniform. Again, please tell me how to decide the metastatic niche. Thank you for this comment. With the goal to clarify this statement, we added a text in the 2.9.2 section. “Metastatic niches in the liver and lung were assessed semi quantitatively as described (Van den Eynden et al., 2012 and Cavalcante et al., 2021). Percentage of tumour cells in the tissue parenchyma represented by Scores was applied (1, <5%; 2, 5% to 25%; 3, 26% to 50%; 4, 51% to 75%; 5, >75%). The scores were applied to histological images captured in 20 random fields of all the extension the organs. Histopathological analysis of tumour tissue, liver, and lungs were independently assessed by two non‐ operator analysts (RA and RC). Three histological sections per animal tissue (n = 5) were analysed in each group”.
In the figure 5 B and 5 D, the metastatic niches are scattered around portal veins in the liver as well as around blood vessels in the lungs as seen in H&E stained histological sections. As the orthotopic model of breast cancer lasts around 21 days, large metastatic niches are not formed and therefore deviations are not as high than with larger tumors. Metastatic niches are small clusters of cells located in the secondary organ. This process occurs due the fact that a long time to the experiment for more three weeks could compromise the animal's systemic health. As 4T1 cells are quite aggressive, a time period greater than 21 days could put animal welfare at risk, which is not allowed in the Helsinki animal experimentation regulations (247/1996, AWA).
Indeed, we investigated on additional marker in the lungs and liver by immunohistochemistry related to immunosuppression and metastasis which is now incorporated in figures 5E-G. CXCL12 is expressed in normal epithelial cells in intestine and mammary glands. Breast cancer is an example for a tumour with an organ-specific pattern of distant metastasis formation. It mainly colonizes lung, liver, lymph nodes and bone marrow, all of which are abundant sources of chemokine ligands a. Overexpression of chemokines - especially of CXCR4 and CXCL12 - was observed in breast cancer cells and surgical specimens, but chemokine receptors are also highly expressed in other tumour types including cancers of epithelial, mesenchymal and hematopoietic origin [b]. The role of CXCL12 in the metastatic cascade of breast cancer and also its ability to predict patient survival have been intensively studied. Several groups found that CXCL12 can promote tumour cell migration and invasion (c).
A) Ali S, Lazennec GChemokines novel targets for breast cancer metastasis. Cancer Metastasis Rev 2007 26: 401–420.
b) CXCR4/CXCL12 Participate in Extravasation of Metastasizing Breast Cancer Cells within the Liver in a Rat Model
C) Mirisola V, Zuccarino A, Bachmeier BE, Sormani MP, Falter J, et al. CXCL12/SDF1 expression by breast cancers is an independent prognostic marker of disease-free and overall survival. Eur J Cancer.2009 45: 2579–2587.
・Then, what is the component of EVs of M1? What is the difference of EVs from M1 vs M2 macrophages? You only showed the phenomena, and I am very interesting, but please discuss or speculate the differences of components. For example, EVs-M1 suppress MMP-2 -> High expression of TIMPs?
Although there is now much information about the internal components of M1-EVs, it is known that EVs possess similar characteristics of their parent cells. As the EVs carry diverse RNA and protein cargos from their parental cells, the composition of exosomes from the activated macrophages has been reported. It has been described that a certain pool of RNA transcripts reflecting the phenotype of M1 or M2 macrophages was specifically sorted and packaged into EVs (A). Analysis of other authors have shown that M1 EVs can upregulate cytokines expression such as iNOS, IL-6, and IL-12 and downregulating expression of anti-inflammatory IL-4 and IL-10. They have also observed mRNA expression of M1- macrophage markers while the opposite effect has been reported by M2 (B). A) Cheng, L.; Wang, Y.; Huang, L. Exosomes from M1-Polarized Macrophages Potentiate the Cancer Vaccine by Creating a Pro-inflammatory Microenvironment in the Lymph Node. Mol Ther 2017, 25, 1665-1675, doi:10.1016/j.ymthe.2017.02.007.
B) Wang, P.; Wang, H.; Huang, Q.; Peng, C.; Yao, L.; Chen, H.; Qiu, Z.; Wu, Y.; Wang, L.; Chen, W. Exosomes from M1-Polarized Macrophages Enhance Paclitaxel Antitumor Activity by Activating Macrophages-Mediated Inflammation. Theranostics 2019, 9, 1714-1727, doi:10.7150/thno.30716.

Round 2
Reviewer 1 Report
The authors have addressed all my comments and have delivered a significantly improved version of the manuscript.
Two minor issues should be addressed before publication:
- Section 3.3. Both in the text and in the legend of Figure 3B, CD63 is mentioned as marker of M1-polarization. Do the authors mean CD63 or CD68? To the best of my knowledge, CD68, which is actually introduced by the authors in materials and methods (section 2.5) can be used as marker of M1 macrophages. CD63, on the contrary, is a tetraspanin frequently used as marker of EVs in general.
- The legend of Figure 4F and G needs to be corrected. Fig 4F corresponds to NFkB and 4G corresponds to CD8.
Author Response
- Section 3.3. Both in the text and in the legend of Figure 3B, CD63 is mentioned as marker of M1-polarization. Do the authors mean CD63 or CD68? To the best of my knowledge, CD68, which is actually introduced by the authors in materials and methods (section 2.5) can be used as marker of M1 macrophages. CD63, on the contrary, is a tetraspanin frequently used as marker of EVs in general.
Dear reviewer, thanks for the comments and all positive feedback to improve our manuscript. You are correct in both statements. We used CD68 as a marker as described in the methodology. We corrected this sentence in the text and figure legend.
- The legend of Figure 4F and G needs to be corrected. Fig 4F corresponds to NFkB and 4G corresponds to CD8.
Thanks for spotting this error. We have corrected it as indicated
